

# Deep water hydrodynamic observations around a Cold-Water Coral habitat in a submarine canyon in the Eastern Ligurian Sea (Mediterranean Sea)

Tiziana Ciuffardi[1], Zoi Kokkini[2], Maristella Berta[2], Marina Locritani[3], Andrea Bordone[1], Ivana Delbono[1], Mireno Borghini[2], Maurizio Demarte[4], Roberta Ivaldi[4], Federica Pannacciulli[1], Anna Vetrano[2], Davide Marini[5], Giovanni Caprino[5]

[1] ENEA - Centro Ricerche Ambiente Marino Santa Teresa, Via S. Teresa, 19032 Lerici, La Spezia, Italy

[2] Consiglio Nazionale delle Ricerche- Istituto di Scienze Marine (CNR-ISMAR) Sede Secondaria di Lerici, Forte Santa Teresa s.n.c, 19032, Lerici, La Spezia , Italy

[3] Istituto Nazionale di Geofisica e Vulcanologia, Via di Vigna Murata 605 00143 Roma, Italy

[4] Istituto Idrografico della Marina, Passo dell'Osservatorio 4, 16134 Genova, Italy

[5] Distretto Ligure delle Tecnologie Marine scrl, Viale Nicolò Fieschi 18, 19123 La Spezia, Italy

*Correspondence to*: Tiziana Ciuffardi (tiziana.ciuffardi@enea.it)

**Abstract.** A 2 years dataset of a stand-alone mooring placed, deployed in November 2020 down the Levante Canyon in the Eastern Ligurian Sea, is presented. The Levante Canyon Mooring (LCM) is a deep submarine multidisciplinary observatory deployed at 608 m depth, in a key ecosystem area. The Levante Canyon hosts a valuable and vulnerable ecosystem of deep-living Cold-Water Corals (CWC), studied and monitored since 2013 through the integrated mapping of seabed and water column. The 2 years dataset, acquired on the mooring and presented here (data from November 2020 to October 2022), includes measurements conducted with both currentmeters and CTD probes, and provides information about the hydrodynamics and thermohaline properties across almost the entire water column. The observatory is still on-going and the dataset is regularly updated. All the described data are publicly available from https://doi.org/10.17882/92236 (Borghini et al., 2022). They must therefore be preserved and are of considerable scientific interest.

## 1 Introduction

The Mediterranean Sea is often seen as an incredible open-air laboratory ideal to study processes and ecosystems at different spatial and temporal scales. It is both a climate-change and biodiversity hotspot, characterised by a high level of marine endemism but also subjected to a constant increase in anthropogenic pressures (e.g. fishing, including deep sea, aquaculture, tourism, exploration and production of oil & gas, coastal development) and their effects (e.g. eutrophication, climate change, proliferation of alien species).



The huge biological complexity characterising marine communities makes the development of ecological monitoring
increasingly indispensable, beyond the traditional time-consuming, high-cost sampling cruises, especially for the deep sea
where less information is available. Furthermore, identifying and quantifying the effect of different anthropogenic stressors
on marine ecosystems requires an integrative and multidisciplinary approach (encompassing the simultaneous measurement
of biogeochemical and oceanographic variables). To this purpose, there are many observational systems and initiatives for
the study and monitoring of the Mediterranean that see Italian institutions as main actors. One of these initiatives, in
particular, is characterised by the collaboration, in terms of human resources, infrastructures and instruments, among: the
Ligurian DLTM (Ligurian District of Marine Technologies acting as coordinator), CNR (National Research Council), ENEA
(Italian National Agency for New Technologies, Energy and Sustainable Economic Development), IIM (Italian Navy -
Hydrographic Institute) and INGV (National Institute of Geophysics and Volcanology). These institutions have developed an
observatory composed of two stations: the first one placed in Smart Bay Santa Teresa (https://smartbaysteresa.com/) at 10 m
of depth and the second one installed in the Levant Canyon in the Eastern Ligurian Sea, the so-called Levante Canyon
Mooring (LCM hereafter).

The coastal station monitors temperature, pressure, water conductivity and derived salinity (Bordone et al., 2022) and its
main purpose is the study of coastal ecosystems, hydrodynamic processes and, in the long term, the effects of climate
change. Moreover, the station hosts an experiment of plastic and bioplastic degradation in marine environments (De Monte
et al., 2022).

The Eastern Ligurian Sea is an extremely interesting area from different aspects: its continental platform has a reduced
extension and constitutes an exclusive marine observation point, not only because it is located inside the "Pelagos
Sanctuary", an area with a high concentration of cetaceans, but also for the presence of the submarine Levante Canyon, an
underwater canyon off Cinque Terre, almost parallel to the coast, which, due to the strong bottom currents and the
considerable contribution of sediments and organic substances, creates an environment favourable to the development and
growth of valuable ecosystems such as deep corals.

The installation of the LCM was aimed to investigate this peculiar deep-sea area where a joint study by ENEA and IIM
carried-out in 2013-2014, with the support of a ROV (Remotely Operated Vehicle), highlighted the presence of living
colonies of *Madrepora oculata* (Cold-Water Corals, CWC) at a depth of about 570 m (see Figure 1 and Delbono et al., 2014;
Fanelli et al., 2017; Pratellesi et al., 2014). These organisms give rise to deep ecosystems with high biodiversity, but suffer a
strong impact caused by trawling. It is well documented that these kinds of areas should be prioritised in monitoring (e.g.,
Canals et al., 2006, Thurber et al., 2014), as they play a fundamental role in shelf-slope connectivity and in the ecological
status of continental margins.

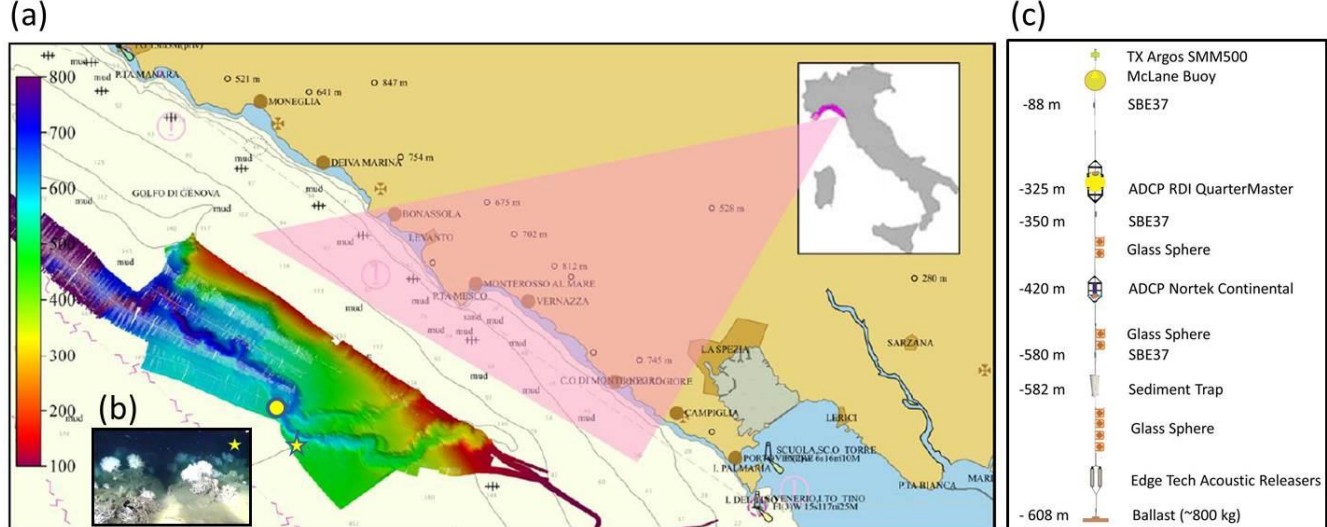

Figure 1: Study Area. panel (a) represents the Eastern Ligurian Sea with the superimposed 2D view of the Levante Canyon. The yellow circle represents the mooring while the yellow star is positioned at the point where dense populations of living, 1 m high colonies of Madrepora oculata were found by ENEA and IIM in 2014 (a detail is reported in panel (b)). The panel (c) represents the mooring layout (not in scale).

The coastal dynamic of the Ligurian Sea is characterised by an East-West cyclonic flux composed of waters arriving from both sides of Corsica island: warm waters coming from the Tyrrhenian Sea through Corsica channel and cold waters rising from the Western Mediterranean current. Merging together at the North of Corsica, these two circular fluxes originate a current with intermediate properties (Astraldi and Gasparini, 1992). Strong time (mainly seasonal) and spatial variations characterise Eastern Ligurian circulation with complex circulation patterns especially in summer. The area is characterised by several sources of frontogenesis and development of meso-submesoscale instabilities (Ciuffardi et al., 2016). On the one hand, the interaction of coastal and shelf waters with the colder and saltier Northern Current (NC) at basin scale (Astraldi and Gasparini, 1985), on the other hand the interaction of coastal waters with buoyant river output, such as Arno and Magra rivers (Cattaneo Vietti et al. 2010, Schroeder et al. 2012). Wind forcing affects as well the upper layer circulation and therefore the interaction among water masses (Astraldi and Gasparini, 1986, Poulain et al., 2020). The interplay of water masses at different scales generates meso-submesoscale fronts and filaments which have been recognized to play an important role in the surface dispersion and patchiness of various types of tracers (biological, pollutants and marine debris), as well as in the transfer from the surface to the interior ocean. The challenge in observing directly the submesoscale range, due to the high variability in both time and space, recently encouraged a transnational (Italy-France) multiplatform experiment in this same area under the framework of the JERICO-S3 Project, H2020 EU Programme (work in progress by Berta et al.). In this context multiple observation platforms have been used to span across different scales and to provide comparable measurements (e.g. glider SeaExplorer, CTD probe, Ferrybox, drifter CARTHE). The area of interest is also





monitored by a HF radar network (CNR group, http://radarhf.ismar.cnr.it) providing continuous (hourly) sea surface current maps covering medium ranges (about 40 km) with high resolution (approx. 1.5 km).

Despite a few studies focused on the variability in the area at short time-spatial scales (Locritani et al. 2010, Schroeder et al. 2012, Berta et al. 2020, Poulain et al. 2020), there are no studies, to our knowledge, focusing also on deep hydrodynamics, due to lack of targeted observations in the area. The extended time-series presented in this work aims to fill this gap. Data were continuously collected between November 2020 and October 2022 and provide unique observations about hydrodynamic processes of the site.

## 2 Data and methods

Data comes from the LCM, a shared infrastructure located along the Levante Canyon in the Eastern Ligurian Sea (Figure 1a). The mooring is placed at around 600 m depth in the Levante Canyon, offshore the Cinque Terre Marine Protected Area at 44°05.443'N, 9°29.900'E (Ciuffardi et al. 2020). It was firstly deployed in October 2019, but the instrumented line accidentally detached from the mooring site and drifted toward the French coast where instruments were collected to be finally definitely re-deployed in fall 2020. LCM is a stand-alone offshore mooring, dedicated to the long term monitoring of hydrological properties of water masses. The installation and maintenance operations of the LCM are carried out thanks to the CNR Research Vessel "Dallaporta" and to the "Leonardo" operated by the Italian Navy, allowing the deep observatory to be positioned at about 6.5 nautical miles off the coast. The mooring operates in delayed-mode and is equipped with sensors that measure physical and biogeochemical parameters along the water column from 83 m to 580 m. Starting from the bottom (Figure 1c), the offshore monitoring station is equipped with a sediment trap placed at a depth of 582 m, which provides information on the supply of sediments and nutrients from the surface to the seabed. Further above, CTD probes are placed at three different depths, respectively at 579 m, 335 m and 85 m. The LCM also includes two Acoustic Doppler Current Profilers (ADCP), placed respectively at 406 m and 325 m depth, that measure currents throughout the water column, in order to monitor the link between near-surface waters and deep ones. The 530 m long mooring scheme is represented in Figure 1c.

Both the ADCP systems measure the intensity and directions of currents along the water column and have a temperature sensor in their transducer head. The CTD probe provides measurements of temperature and salinity (along with pressure). The measurements span from 2019 until 2022. Acoustic releases guarantee the recovery, by bringing the entire instrumented line back to the surface during ordinary or extraordinary maintenance. LCM, still operative, is configured and maintained for continuous long-term monitoring. Ordinary maintenance operations are planned every 6-8 months, when the whole structure is recovered for instrumentation check, data downloading, and maintenance (e.g. changing batteries and sediment trap bottles, cleaning, sensors' calibration or substitution of components).

The upward-looking ADCP used is an RDI QuarterMaster (Teledyne RD Instruments USA, Poway, California), using a four-beam, convex configuration with a beam angle of 20° and a working frequency of 153.6 kHz. The instrument is moored




at a mean nominal depth of 325 m with a number of depth cells set to 44, a cell size of 8 m and a blanking distance of 3.5 m. Velocity accuracy is ±1 %, ±5 mm/s. The temperature sensor has the following characteristics: range -5° to 45°C, precision ±0.4°C, resolution 0.01°.

The downward-looking ADCP used is a Nortek Continental (Nortek AS, Norway), using a three-beam, convex configuration with a beam angle of 25° and a working frequency of 190 kHz. The instrument is moored at a mean nominal depth of 420 m with a number of depth cells set to 30 and a cell size of 8 m and a blanking distance of 6.1 m. Velocity accuracy is ±1 % of the measured value. The temperature sensor has the following characteristics: range -4° to 40°C, precision 0.1°C, resolution 0.01°.

The sampling interval is 2h for the RDI and 1h for the Nortek. The sound speed is computed by the pressure and temperature sensors embedded at the transducer head joint to the assumed salinity.

The three SBE37 CTDs deployed have the following characteristics for conductivity and temperature sensors respectively: range 0 to 70 mS/cm and -5 to 35 °C, accuracy 0.003 mS/cm and 0.002 °C, resolution 0.0001 mS/cm and 0.0001 °C. The probes are also equipped with a Strain-gauge pressure sensor with an accuracy of ± 0.1% of the full-scale range.

## 2.1 Dataset and Metadata description

The collection is composed of 5 datasets containing observational data and related metadata from the LCM mooring site for the period November 2020 – October 2022. Eight files, respectively two for ADCP data (in NetCDF format) and six for CTD data (three files respectively in ODV and CSV format), have been submitted, and each file description specifies the probe and its depth (Borghini et al., 2022).

The Metadata report Dataset Information (DI) contains a brief summary description of the dataset and details about their geospatial position, temporal extension and data interval, the institution responsible for measurements, principal investigator name and contact, the observational network to which the mooring belongs, the keywords vocabulary used. The Variables in Dataset (VD) contain specific information about the data structure and variables. The ADCP dataset provides details about the station name, the geographical position, the time coverage, the bottom depth, the cell depth/range and the current components. The ADCP and CTD variables in the dataset are reported as quality controlled as results of QC procedure reported in the next chapter. The headers of filtered ADCP variables are followed by the suffix "_QC" and the appropriate flags by the suffix "_QC_flag".

## 2.2 Data quality check

All data from LCM, after maintenance operations, are subjected to a quality validation system according to international protocols and standards (IOC-UNESCO).



A first visual check of CTD data time series was firstly applied in order to detect spikes and anomalous values. This was made possible after a statistical analysis of our dataset on the property (i.e. temperature and salinity gradient) distribution and frequency to identify the proper thresholds. After these quality checks, the CTD at 88 m has been cut off in the plots at mid-

march 2022 disregarding Conductivity data (and consequently, Salinity and Density) that were affected by fouling artefacts or calibration issues. Temperature data at 88 m were assumed as reliable up to the beginning of August 2022. The same applies for the CTD at 350 m depth that ends in June 2022 in the plots regarding Salinity data. To avoid similar issues in future deployments, the probe drift will be regularly verified and calibrated by conducting vertical profiles using a SeaBird SBE 19 plus calibrated probe.

For the RDI ADCP, the adopted QC procedure was based principally on the 'Manual for Real-Time Quality Control of In-Situ Current Observations' by IOOS (2019), along with the Crout and Conlee (2006) report by NOAA. Four groups of tests have been applied: a) the sensors' overall health, including tests for the sensors' tilt and speed of sound control. Pressure sensors extremes for in & out of water check. Along with the sensors' overall health, a temperature despiking test is also applied, b) the signal quality test that controls the quality of the Transmitting/Receiving signal and includes the Correlation

Magnitude (CM) test and the Percentage Good (PG) ratio, c) the current velocity tests that ensure the validity of the measured current and include the Horizontal and Vertical velocity control and the Error velocity test, and d) the overall profile tests, that control the Echo amplitude (intensity) test.

The analysis and processing of the NORTEK data are done using the SURGE program provided by NORTEK (https://www.nortekgroup.com/softwarelicense/surge). For the Post-Processing, the configuration is the following: a) The

Sidelobe rejection is set to 90%, and by that, as it is downward oriented, it neglects data near the bottom, b) the Low signal to Noise Ratio (SNR) threshold is set up to 3dB, c) allow removing tilt effects, d) the velocity variation test neglecting data over five standard deviations (std), and e) the echo spikes test limit is set up to 70dB. Also, a temperature despiking test is applied as in the RDI dataset.

Both RDI and the NORTEK ADCP data are post-processed using a first-level Quality Control (QC). Then a set of flags is

used to describe the results of the QC. The OceanSites and Copernicus Marine In Situ flag scales were adapted, simplified, and adjusted for a delayed mode of operation for flagging the data (Copernicus In Situ TAC, 2021). Table 1 explains the flags used in all the ADCP datasets.





**Table 1: Code of data qualifiers flags**

| Flags | Description | Result |
|---|---|---|
| 0 | - | No QC applied |
| 1 | Data have passed critical QC tests and are deemed adequate for use as preliminary data | Pass |
| 2 | Data are considered suspect or highly interesting to operators and users. They are flagged suspects to draw further attention to their operators | Suspicious data or high-interest data |
| 3 | Potentially correctable insufficient data. These data cannot be used without scientific correction or re-calibration | Potentially correctable bad data These data are not to be used without scientific correction or re-calibration |
| 4 | Data are considered to have failed one (or more) critical QC checks. If they are disseminated at all, it should be readily apparent that they are not of acceptable quality | Fail |
| 5 | Value changed | Not Used |
| 6 | Value below detection | Not Used |
| 7 | Nominal values | Data were not observed but reported |
| 8 | Interpolated values | Missing data may be interpolated (Not used in our case) |
| 9 | Data are missing; used as a placeholder | Missing data |



## 3. Results

The first two-year dataset of the LCM is proposed here. Seasonal variability is well shown as well the necessity to have
regular maintenance operations to have proper salinity measurements.

Observed trends will be soon tested and integrated by sediment traps sampling. Geochemical information will be then directly related to multiparametric environmental information obtained via the simultaneous collection of oceanographic data.

### 3.1 Thermohaline records

Temperature records measured by the RDI ADCP (at 325 depth) and by the three CTD (at 88, 350 and 580 m depth) are presented here to compare data at different depths along the water column (Figure 2). The data are presented with a 3-day smoothing window. Temperature data recorded at 420 m depth by the Nortek ADCP have been disregarded as being too high due to electronic issues. As a fact, the instrument batteries discharged earlier than scheduled due to a probable overheating. On the contrary, it is interesting noting that the two time series at 325 (ADCP) and 350 m (CTD) depth have synchronous
fluctuations and slight differences. Interestingly the measurements cover the period during the spring-summer 2022, characterised by the exceptional heat wave that began in the second half of April and mainly overheated the central and north-western part of the Mediterranean. This heat wave is not so evident from our temperature records.

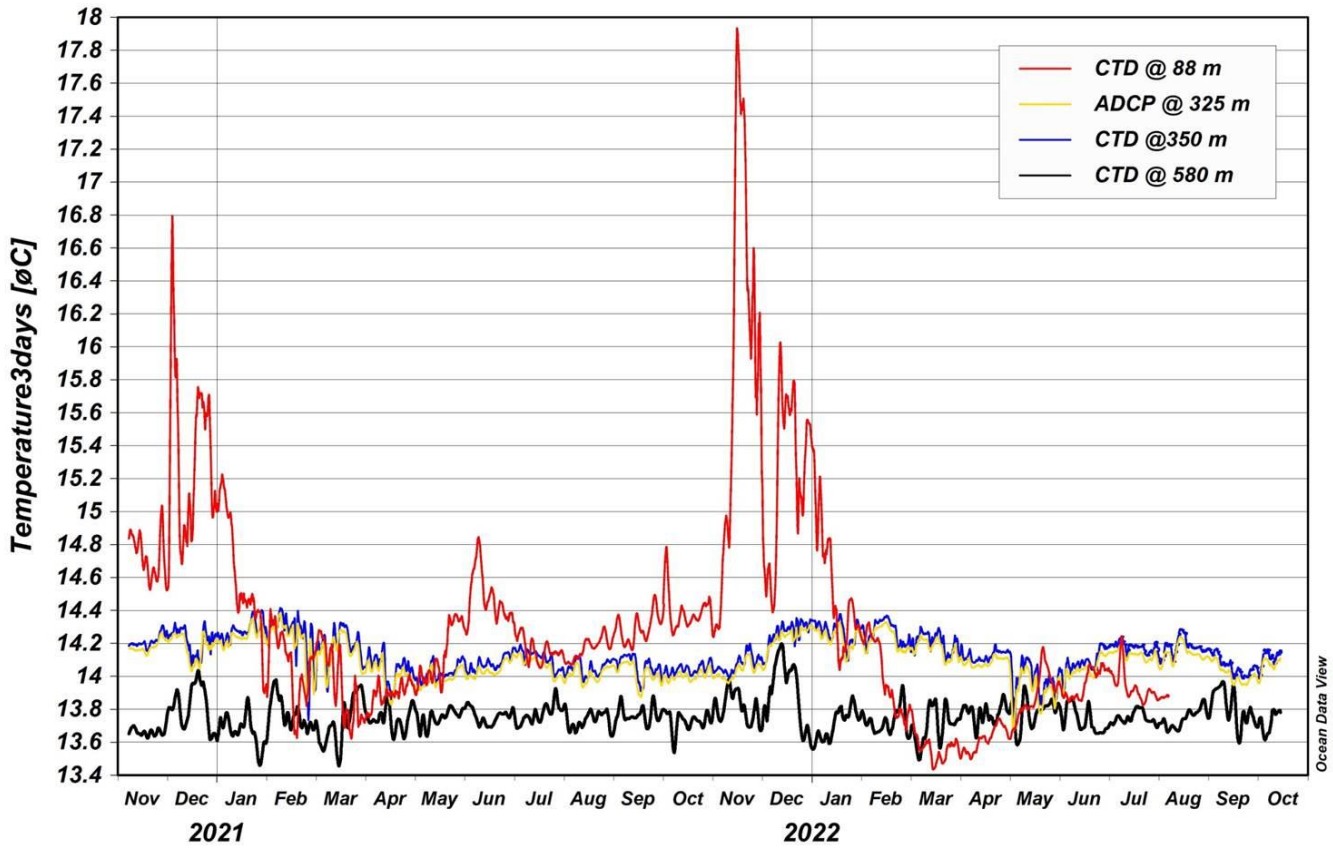

**Figure 2: CTD temperature records on mooring site. The data are presented with a 3-day smoothing window.**

In the upper layer of the Levante Canyon the temperature recorded by the CTD at 88 m has a mean value of 14.33+/-0.35°C
with a minimum temperature of 13.44°C in March 2022 and maximum of 17.93°C in November 2021. At 325 m the ADCP
highlighted an average temperature of 14.10+/-0.11°C with a minimum temperature of 13.67°C and maximum of 14.39°C in
February 2021, in accordance with the nearby CTD positioned at 350 m depth (mean temperature of 14.14+/-0.11°C,
minimum temperature of 13.68°C and maximum of 14.42°C in February 2021). The mean difference between temperature
measured by ADCP and CTD is in the range 0.03-0.05°C (Table 2). In the deepest part of the mooring, near the bottom, at
580 m depth, the mean recorded temperature is 13.76+/-0.10°C with a minimum temperature of 13.46°C and maximum of
14.20°C in December 2021.

The first CTD (from top, in the mooring layout) at 88 m depth well represents the seasonal cycle in the first hundred metres
of the water column, where the mixed layer depth varies according to the seasonal thermocline and temperature changes are
more evident. In particular, the temperatures experienced high peaks during the autumn-early winter months: this is the
result of a periodic annual cycle where surface warmer waters, after the summer heating, experienced strong vertical mixing





from November until January, producing a period of vertical homogeneity of the surface layers (at least up to 88 m according to the measurements). Wind episodes from late summer through fall contribute to this vertical mixing. On the contrary, solar radiation in spring and summer periods results in the development of water column stratification with greater

differences in the annual temperature variation at the sea surface compared with that in the underlying layers (at 88 m). The only exception looks like early summer 2021 when temperatures in the upper layer still oscillate, proving some inter-annual variability. This trend is also evident from the Hovmöller diagrams of temperature and salinity (Figure 3) that provide a synthetic view about the dynamics across the water column during the time. To highlight them, in Figure 3, we display the vertical sections associated with the three layers from 88 to 580 m, obtained by applying the DIVA software tool (Data-

Interpolating Variational Analysis) along all the available data. It shows the temporal sequence resulting from November 2020 to October 2022 across the canyon, considering both the CTD and ADCP measurements as regards temperature records around 325-350 m depth. From this diagram red and orange stripes represent the temperature peaks during winter 2020 and 2021, covering the upper water column around 88 m water depth. On the contrary, the intermediate and the deeper layers show less oscillations, even if some variability and colder periods are present (i.e. vertical thick blue lines).

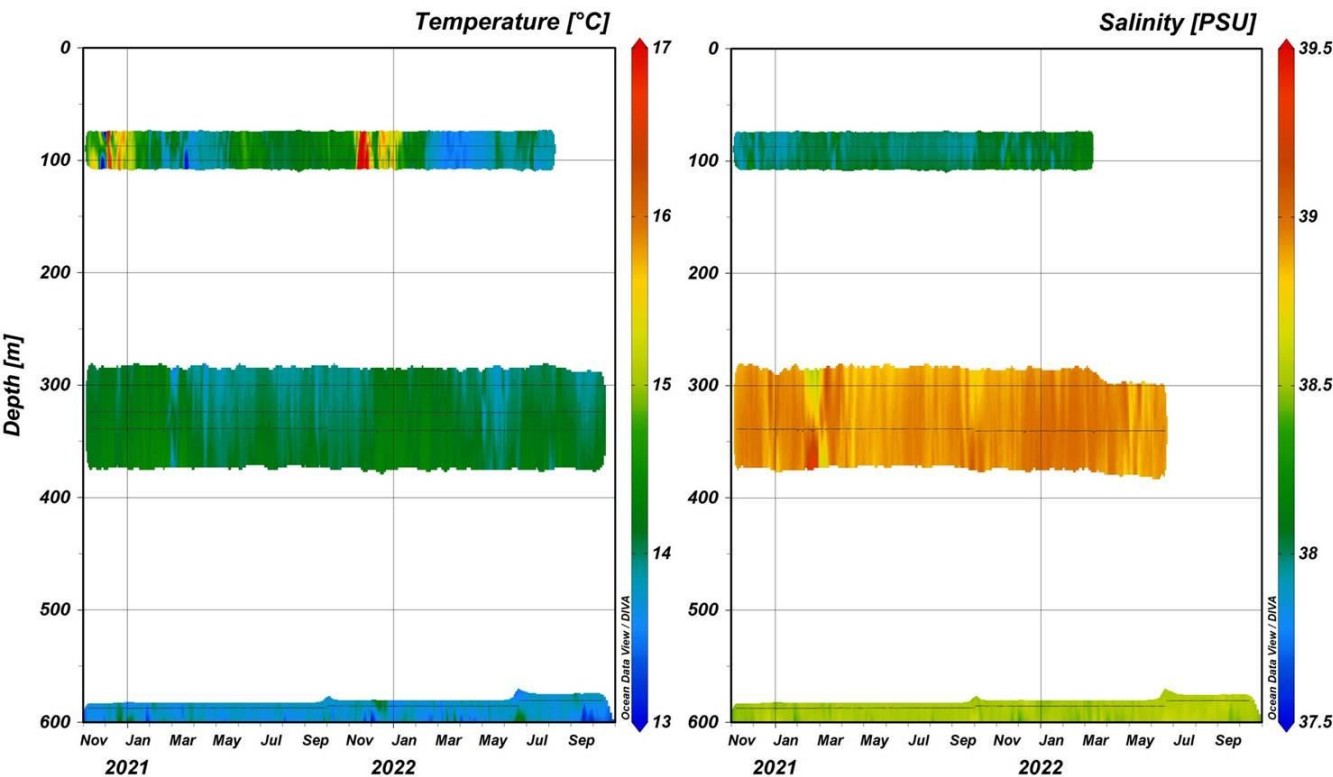


**Figure 3: Hovmöller diagrams of temperature and salinity records for the three layers between 88 and 580 m depth, obtained by applying the DIVA software tool along all the available CTD and ADCP measurements.**

As regards salinity measurements, the latest data from the CTD probes at 88 and 350 m were not plotted as they failed QC due to calibration issues. Hence the data at 88 m depth ends in February 2022 in Figure 4, while the latest data at 350 m
depth goes up to June 2022. The salinity is lower in the upper layer of the Levante Canyon, where the salinity recorded by the CTD at 88 m has a mean value of 38.01+/-0.05 PSU with a minimum of 37.87 PSU and maximum of 38.17 PSU. In the central part of the mooring, at 350 m the CTD recorded the higher salinities, with an average value of 38.95+/-0.04 PSU with a minimum of 38.73 PSU and maximum of 39.03 PSU. Near the bottom, at 580 m depth, the time-series of salinity shows intermediate values with respect to upper and deeper parts: the mean recorded salinity is 38.54+/-0.03 PSU with a minimum
of 38.48 PSU and maximum of 38.61 PSU. Again, the Hovmöller diagram shows larger oscillations for the upper layer at 88 m depth, while salinity values are less variable, especially for the deepest part of the canyon (Figure 3).

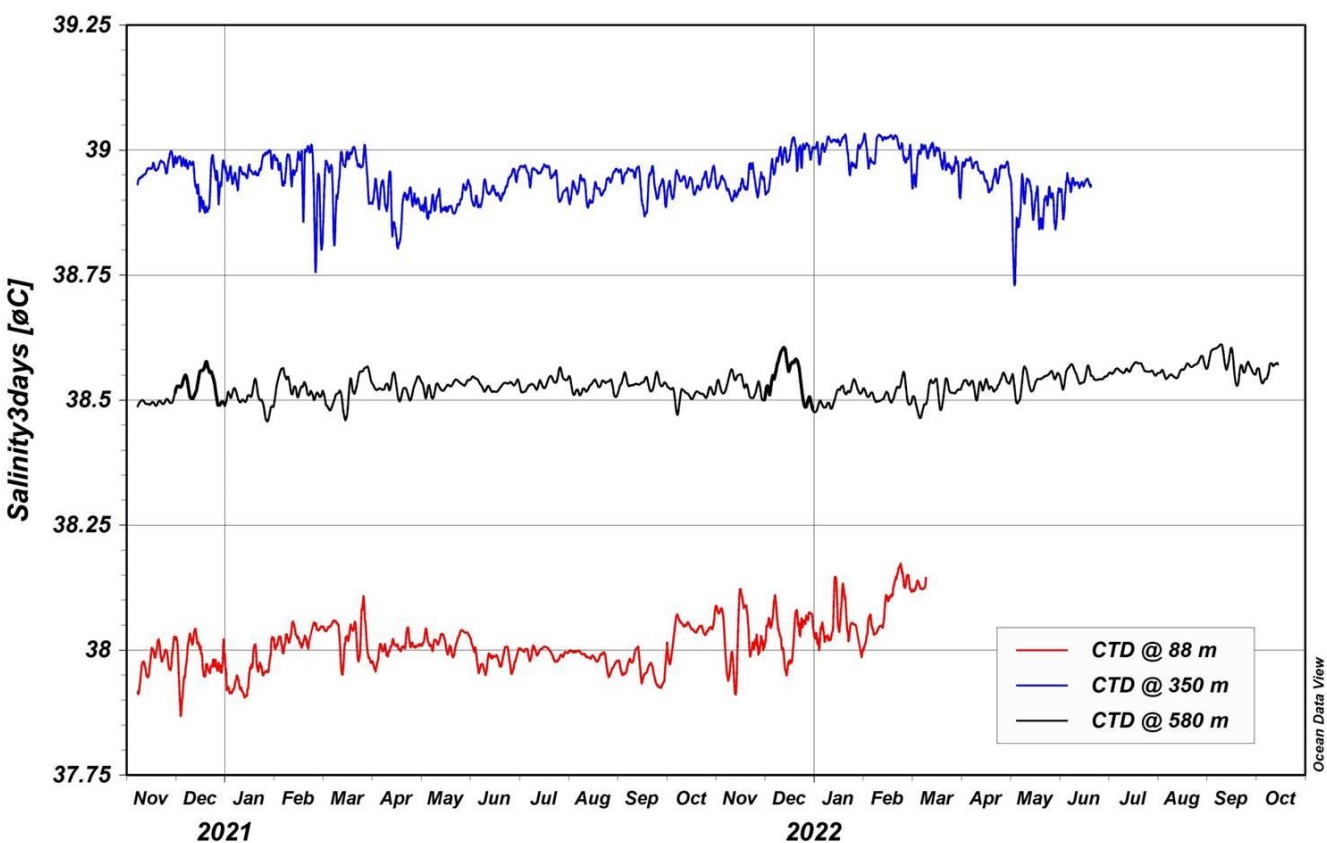

**Figure 4: CTD salinity records on mooring site. The data are presented with a 3-day smoothing window.**

The potential temperature vs. salinity (T S) diagram of the whole data set is shown in Figure 5. The large variability at 88m
indicates the complex water mass structure subjected to strong seasonal influence of heat and water exchanges with the atmosphere, while below 325 m there is a tight correlation indicating relatively unchanging water masses at these depths in

the region, with values in the ranges 13.7–14.4°C/13.3-14.1°C and 38.7–39.1/38.4-38.6 PSU, at 350 and 580 m depth respectively. These values are consistent with the characteristics of the surface water of Atlantic origin (Atlantic water, AW; upper 150 m) and of the Levantine Intermediate Water (LIW; from about 200 to 700 m depth) recorded for the western

Mediterranean (Fedele et al., 2022; Iacono et al. 2021) and Ligurian Sea (Margirier et al., 2020; Prieur et al. 2020). Statistics about temperature and salinity records grouped by months are reported in Table 2 and Table 3.

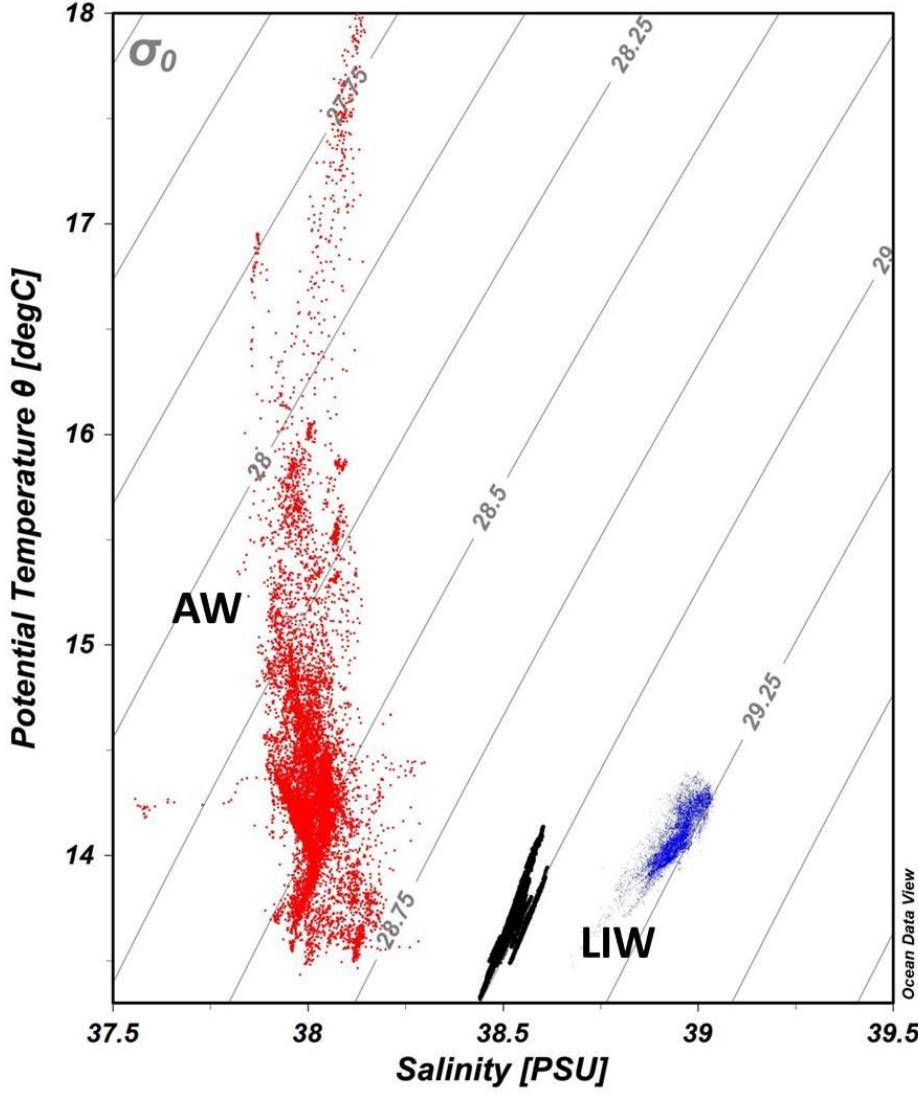

**Figure 5: Temperature-salinity (T-S) diagram for the Levante Canyon Site, obtained from CTD measurements covering the period November 2020– October 2022. Regions occupied by the Atlantic Water (AW) and the Levantine Intermediate Water (LIW) are**
**highlighted.**





**Table 2: Statistical parameters of the 3-days averaged temperatures vs. months. SD stands for Standard Deviation. Both ADCP and CTD data are reported for the depth 325/350 m respectively. The mean difference between temperature measured by ADCP and CTD is in the range 0.033-0.051°C.**

| Month | CTD @ 88 m | | | | ADCP/CTD @ 325/350 m | | | | | | | | CTD @ 580 m | | | |
|---|---|---|---|---|---|---|---|---|---|---|---|---|---|---|---|---|
| | Mean | SD | Min | Max | Mean (°C) | | SD | | Min (°C) | | Max (°C) | | Mean (°C) | SD | Min (°C) | Max (°C) |
| | | | | | CTD | ADCP RDI | CTD | ADCP RDI | CTD | ADCP RDI | CTD | ADCP RDI | | | | |
| | (°C) | | | | | | (°C) | | | | | | (°C) | | | |
| January | 14.585 | 0.356 | 13.898 | 15.399 | 14.278 | 14.243 | 0.057 | 0.053 | 14.094 | 14.088 | 14.401 | 14.358 | 13.693 | 0.092 | 13.461 | 13.882 |
| February | 14.131 | 0.153 | 13.625 | 14.408 | 14.266 | 14.227 | 0.106 | 0.099 | 13.735 | 13.849 | 14.415 | 14.385 | 13.751 | 0.086 | 13.623 | 13.980 |
| March | 14.052 | 0.167 | 13.435 | 14.280 | 14.197 | 14.161 | 0.084 | 0.077 | 13.907 | 13.891 | 14.327 | 14.282 | 13.705 | 0.121 | 13.457 | 13.949 |
| April | 13.894 | 0.064 | 13.497 | 14.009 | 14.088 | 14.047 | 0.069 | 0.068 | 13.860 | 13.829 | 14.204 | 14.142 | 13.759 | 0.062 | 13.618 | 13.897 |
| May | 14.105 | 0.164 | 13.619 | 14.379 | 14.002 | 13.951 | 0.076 | 0.081 | 13.676 | 13.668 | 14.161 | 14.113 | 13.767 | 0.070 | 13.585 | 13.943 |
| June | 14.477 | 0.152 | 13.843 | 14.845 | 14.078 | 14.038 | 0.076 | 0.049 | 13.676 | 13.890 | 14.161 | 14.150 | 13.750 | 0.061 | 13.654 | 13.865 |
| July | 14.173 | 0.077 | 13.826 | 14.384 | 14.148 | 14.107 | 0.050 | 0.047 | 13.997 | 13.969 | 14.242 | 14.190 | 13.761 | 0.048 | 13.686 | 13.928 |
| August | 14.165 | 0.062 | 13.855 | 14.302 | 14.112 | 14.073 | 0.080 | 0.077 | 13.965 | 13.932 | 14.286 | 14.240 | 13.742 | 0.052 | 13.638 | 13.862 |
| September | 14.292 | 0.083 | 14.159 | 14.494 | 14.066 | 14.020 | 0.059 | 0.057 | 13.910 | 13.874 | 14.169 | 14.120 | 13.788 | 0.083 | 13.596 | 13.969 |
| October | 14.398 | 0.112 | 14.248 | 14.786 | 14.062 | 14.022 | 0.048 | 0.043 | 13.967 | 13.950 | 14.164 | 14.110 | 13.725 | 0.059 | 13.537 | 13.801 |
| November | 14.636 | 0.337 | 14.238 | 17.934 | 14.134 | 14.097 | 0.083 | 0.085 | 13.993 | 13.958 | 14.309 | 14.268 | 13.749 | 0.100 | 13.626 | 13.983 |
| December | 14.920 | 0.280 | 14.388 | 16.793 | 14.237 | 14.204 | 0.078 | 0.074 | 14.033 | 14.030 | 14.353 | 14.312 | 13.869 | 0.161 | 13.590 | 14.199 |

**Table 3: Statistical parameters of the 3-days averaged salinity data vs. months. SD stands for Standard Deviation.**

| Month | CTD @ 88 m (PSU) | | | | CTD @ 350 m | | | | CTD @ 580 m | | | |
|---|---|---|---|---|---|---|---|---|---|---|---|---|
| | Mean | SD | Min | Max | Mean | SD | Min | Max | Mean | SD | Min | Max |
| | (PSU) | | | | (PSU) | | | | (PSU) | | | |
| January | 37.998 | 0.061 | 37.890 | 38.147 | 38.982 | 0.029 | 38.919 | 39.031 | 38.504 | 0.019 | 38.457 | 38.543 |
| February | 38.040 | 0.029 | 37.994 | 38.173 | 38.980 | 0.050 | 38.756 | 39.033 | 38.519 | 0.018 | 38.491 | 38.564 |
| March | 38.034 | 0.031 | 37.951 | 38.145 | 38.967 | 0.044 | 38.801 | 39.014 | 38.512 | 0.027 | 38.460 | 38.567 |
| April | 38.002 | 0.017 | 37.957 | 38.045 | 38.925 | 0.043 | 38.803 | 38.985 | 38.527 | 0.013 | 38.497 | 38.555 |
| May | 38.017 | 0.014 | 37.985 | 38.043 | 38.890 | 0.035 | 38.730 | 38.973 | 38.532 | 0.016 | 38.493 | 38.567 |
| June | 37.983 | 0.019 | 37.950 | 38.030 | 38.922 | 0.019 | 38.861 | 38.963 | 38.539 | 0.014 | 38.516 | 38.572 |
| July | 37.995 | 0.009 | 37.977 | 38.009 | 38.950 | 0.021 | 38.897 | 38.971 | 38.547 | 0.017 | 38.515 | 38.576 |
| August | 37.985 | 0.014 | 37.946 | 38.000 | 38.923 | 0.018 | 38.885 | 38.962 | 38.544 | 0.020 | 38.506 | 38.588 |
| September | 37.961 | 0.022 | 37.924 | 38.007 | 38.936 | 0.028 | 38.867 | 38.968 | 38.555 | 0.027 | 38.510 | 38.611 |
| October | 38.037 | 0.023 | 37.972 | 38.071 | 38.934 | 0.016 | 38.886 | 38.969 | 38.526 | 0.025 | 38.470 | 38.574 |
| November | 38.026 | 0.054 | 37.911 | 38.122 | 38.946 | 0.024 | 38.898 | 38.998 | 38.514 | 0.021 | 38.487 | 38.561 |
| December | 38.015 | 0.043 | 37.868 | 38.110 | 38.963 | 0.039 | 38.874 | 39.026 | 38.540 | 0.032 | 38.482 | 38.605 |

**3.2 Hydrodynamic records**

This section reports the hydrodynamic data measured by the two ADCPs along the water column in the Levante Canyon site, from 2020 to 2022. For better visualisation, in order to better solve the dynamic variability of the water column, data have

been separated in 5 vertical layers of approximately 100 m each: UL (upper layer) 50-150 m, UIL (upper intermediate layer) 150-250, IL (intermediate layer) 250-350 m, LIL (lower intermediate layer) 400-500 and BL (bottom layer) 500-600 m depth. For each layer, polar scatter diagrams have been plotted, with speed sorted every 0.2 m/s.

Figure 6 shows the 2-years-long ADCP records as vertical distribution of the speed module along the water column, while polar scatterplots are reported in Figure 7, which represent the direction and intensity of currents along the water column.

Current data shows an average weak hydrodynamic field at the BL (0.06+/-0.02 ms-1) able to reach a speed of 0.76 ms-1 at

about 50 m depth during events of strong current. These currents are generally recorded during winter-early spring (between December and April) every year and interest the whole water column. Stronger currents also occur during summer 2021, but remain confined to the UL and UIL layers. The hydrodynamic field of the five selected layers is characterised by currents spreading between 120 and 180°N and between 270-330°N. This behaviour indicates a flow oriented toward the canyon axis
and along the direction of the isobaths (see Figure 1). In the two deeper layers (i.e. LIL and BL) this directional spreading slightly rotates assuming a North-South orientation.

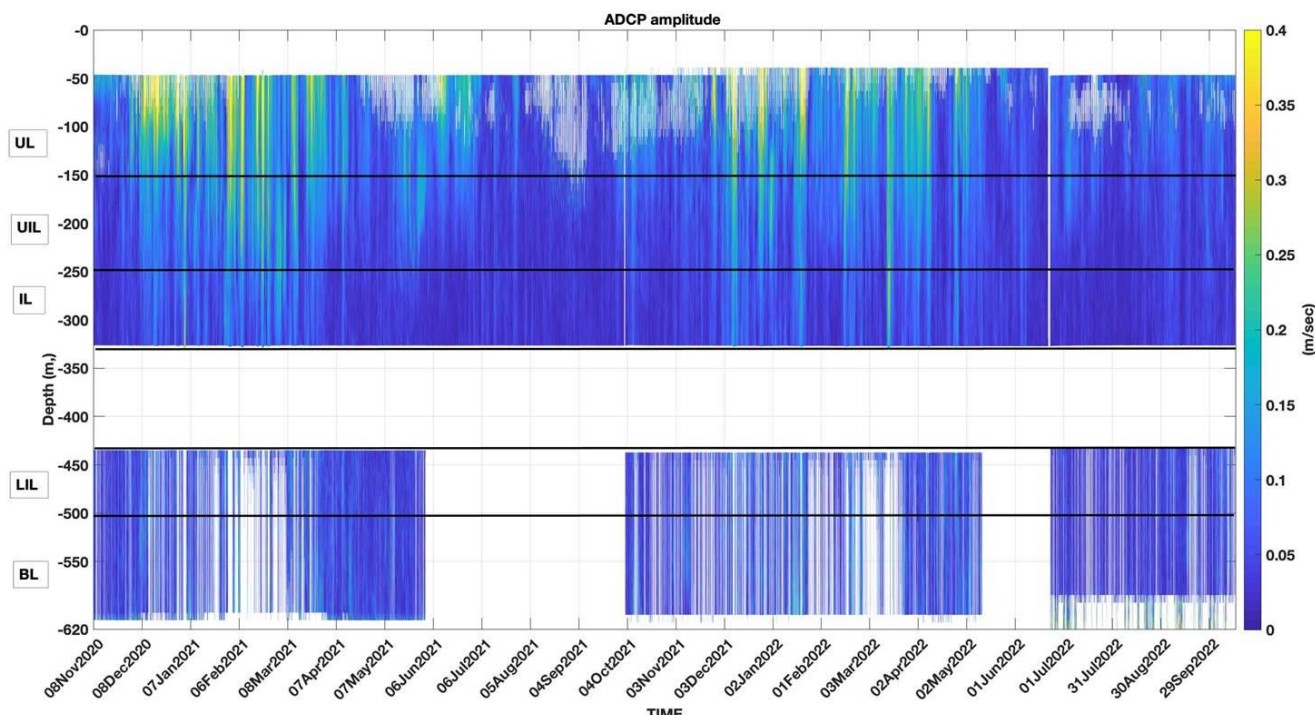

**Figure 6: Currents speed records along the water column (data filtered as described in the data quality check section), boxes on the left and straight lines indicate the 5 different layers assumed in the analysis (UL: Upper Layer; UIL: Upper Intermediate**
**Layer; IL: Intermediate Layer; LIL: lower intermediate layer; BL: Bottom Layer)**

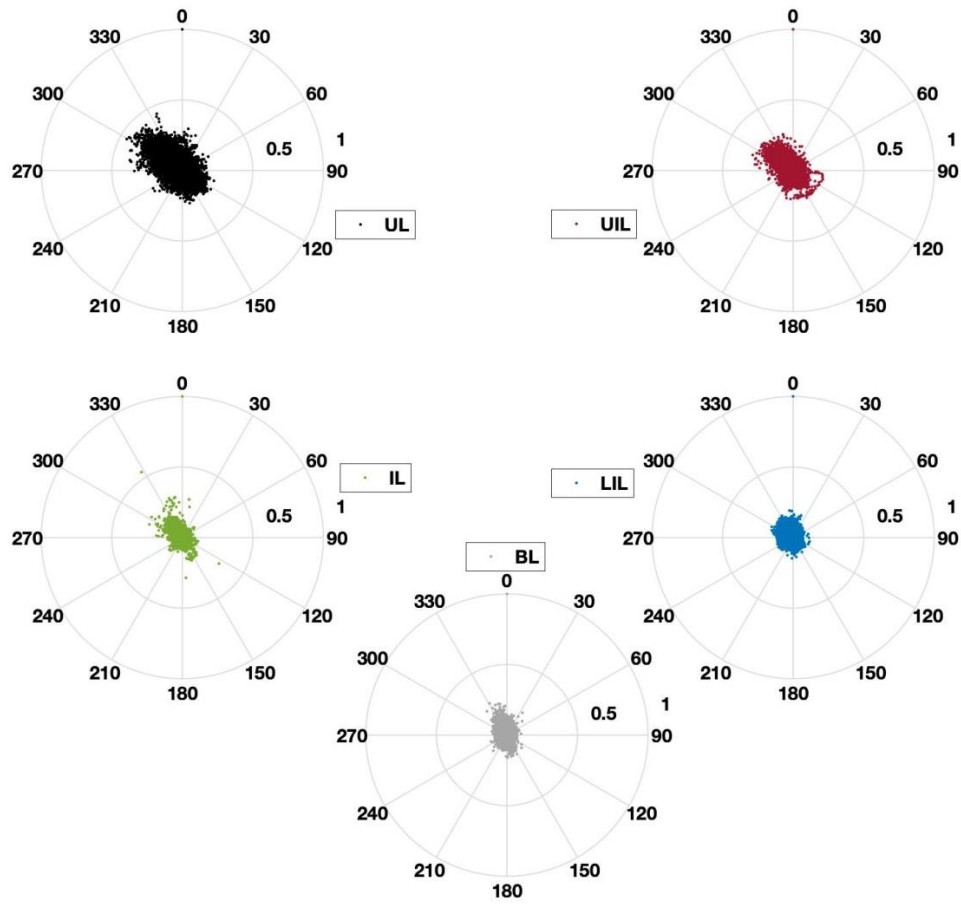

**Figure 7: Polar scatter plot of observed directional current velocity (ms-1). (UL: Upper Layer; UIL: Upper Intermediate Layer; IL: Intermediate Layer; LIL: lower intermediate layer; BL: Bottom Layer).**

Figure 8 shows the time series of the five layers by applying a daily average smoothing. Focusing on the U and V components from the ADCP (Figure 9 and 10), the time series shows a reversal of the U component in summer/early fall both for 2021 and 2022. Especially in June-July, the U component shows significant positive sign (from west to east), while the typical NC negative sign (from east to west) prevails during winter, also associated with larger V magnitude compared to the summer period. The NC episodic and local reversal during summer has been observed as well in surface (1 m depth) by

current maps from the CNR-ISMAR HF radar network covering the area nearby the mooring since 2016 (Berta personal communication). The future combination of these datasets gives the opportunity to further investigate the origin of the reversal, the coherence of the signal throughout the water column (also considering the typical strongly stratified waters profile in summer), and to what extent the current reversal affects the canyon dynamics.

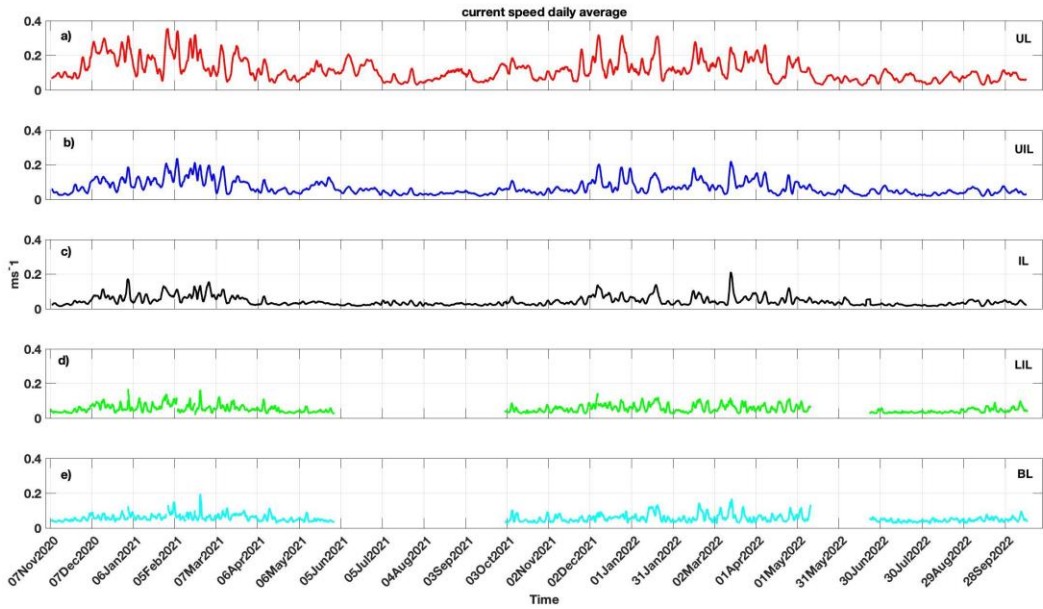

**Figure 8: Time series of the current speed in the 5 different layers assumed in the analysis by applying a daily average smoothing (UL: Upper Layer; UIL: Upper Intermediate Layer; IL: Intermediate Layer; LIL: lower intermediate layer; BL: Bottom Layer).**

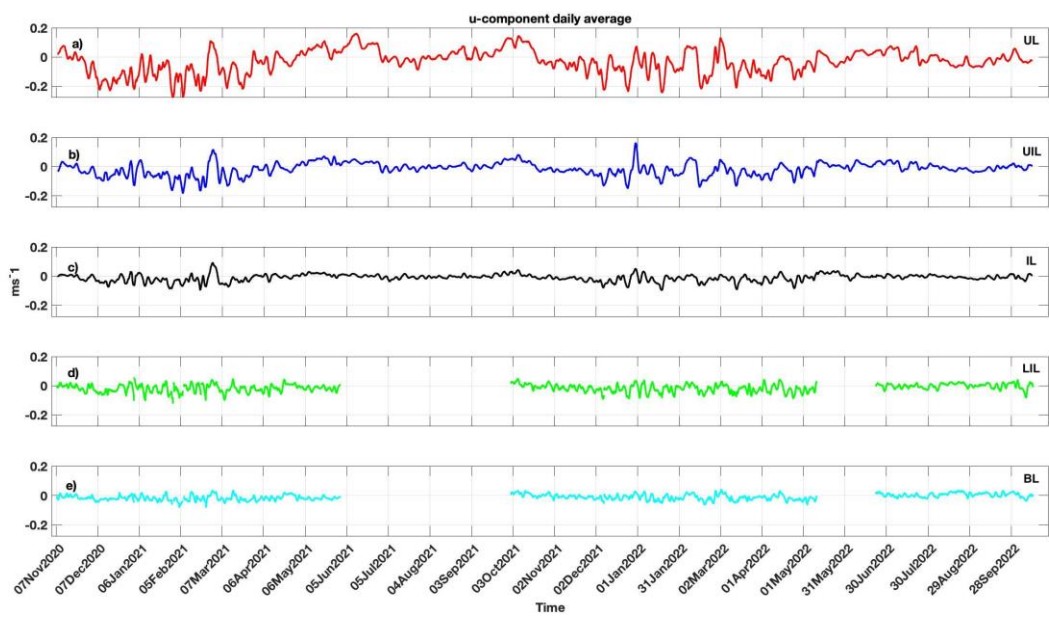

**Figure 9: Time series of the East component of the current speed in the 5 different layers assumed in the analysis by applying a daily average smoothing (UL: Upper Layer; UIL: Upper Intermediate Layer; IL: Intermediate Layer; LIL: lower intermediate layer; BL: Bottom Layer).**



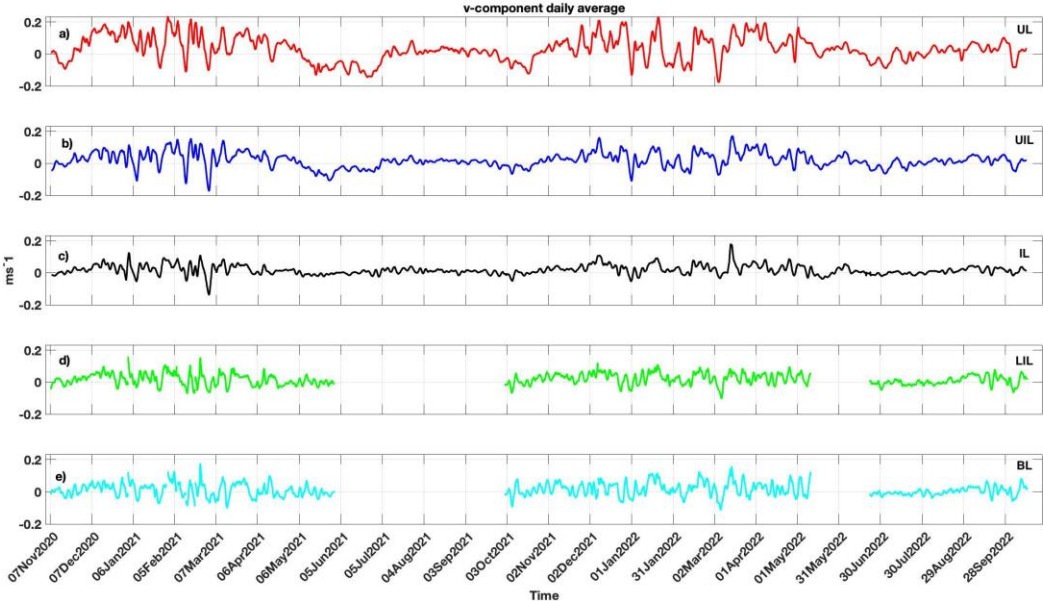

**Figure 10: Time series of the of the North component of the current speed in the 5 different layers assumed in the analysis by applying a daily average smoothing (UL: Upper Layer; UIL: Upper Intermediate Layer; IL: Intermediate Layer; LIL: lower intermediate layer; BL: Bottom Layer).**

## 4. Conclusions

This paper presents the results of the first 2 years of the Levante Canyon Mooring activity conducted in the Eastern Ligurian Sea, throughout the installation in 2020 of a mooring line placed in a canyon area at about 600 m depth, characterised by the presence of cold water living corals. This activity was realised thanks to a multidisciplinary framework, where some of major national research centres (i.e. CNR, DLTM, ENEA, IIM and INGV) cooperate under the coordination of the DLTM.

The LCM, equipped with two ADCP and three CTD probes, provides hydrodynamic and thermohaline measurements along the water column (approximately from 50 m depth up to the bottom). The LCM initiative can represent an innovative regional Centre where members can calibrate, compare and test instrumentations or share tools favouring a sustainable technological development in the marine and maritime transport sectors. Furthermore, it can link with the various regional monitoring systems already in existence in both the Tyrrhenian and Ligurian Sea regions, contributing to create a regional network of experimental marine stations. This will meet national and international demand for the protection of the marine environment and its potential for innovation and growth (Marine Strategy and Blue Growth). Its data can nourish studies about human impact on coastal and port areas to stimulate the implementation of European directives and/or national/regional laws (e.g. ecosystem approach and environmental status evaluation).

310 Analysed data extended from 2020 to 2022 and are characterised by seasonal cycle in the first hundred metres of the water column with strong vertical mixing during autumn and winter against water column stratification in spring and summer. This was also stressed by Picco et al. (2010) in their work on the Ligurian Sea, where they describe a vertical thermal structure in the upper thermocline characterised by a high variability during winter, related to the presence of the Ligurian front, the occurrence of internal waves and the wind mixing.

315 The time series covers the period of summer 2022, characterised by the exceptional heat wave interesting the central and north-western part of the Mediterranean. In this period the ESA-funded project CAREHeat (https://eo4society.esa.int/projects/careheat/) reports that the Ligurian Sea was particularly affected as the peak of the heat wave was reached in its surface waters in late July, with temperatures of five degrees above average, and, despite a slight drop in temperatures at the end of August, the heat wave was still evident in the measurements of September. From our 320 temperature data in the canyon area, this heat wave is not so evident in the upper layer, where the temperatures are lower than those of summer 2021, whereas a slightly positive increment is recorded for the lower CTD records.

From a hydrological point of view, water mass distribution resulting from the T-S diagrams in the canyon area is coherent with previous evidence in the Ligurian Sea: the canyon is characterised by the presence of two main water masses: 1) Surface water of Atlantic origin (Atlantic water, AW; upper 150 m), well separated from the underlying intermediate water, 325 and 2) the Levantine Intermediate Water (LIW) that occupies the layer between 200 m and 500-700 m of depth. Salinity measurements at 350 m depth appear saltier in the area of the canyon, with regards to standard LIW values. This is something that should be accurately verified during next data recovery and probe calibrations, even if already previously documented by Prieur et al. (2020), as regards coastal and frontal zones of the Ligurian Sea.

The first 2-years' time-series is presented here, but further detailed studies and long-term series of geophysical and 330 hydrological data are necessary to better understand the bottom dynamics, the seabed and water column interaction and the ecological conditions of valuable ecosystems in the Levante Canyon and in the Ligurian Sea more in general, a challenging area for geological, geophysical, oceanographic and ecological research.

## 5. Data availability

All data is made publicly available through https://www.seanoe.org/data/00810/92236/. The registered database DOI is 335 https://doi.org/10.17882/92236 (Borghini, et al., 2022).

This paper describes in detail the temporal coverage of the dataset which is constituted by quite continuous high temporal resolution time series of currents, temperature and salinity from November 2020 to October 2022. The adopted methodology about settings, data records and quality control procedures ensure compliance and consistency of the dataset. The dataset





presented here ends in October 2022 but monitoring activities are still in progress and future data collected by this deep
340    observatory will be added to an updated version of the repository at least every 2 years.

## 6. Author contribution

Conceptualization, T.C., Z.K. and M.B.; Field measurements and data management, M.B., A.B., M.D., Z.K.; Data
processing and analysis, T.C. and Z.K.; TC prepared the manuscript with contributions from all co-authors. All authors have
read and agreed to the published version of the manuscript.

345    ## 7. Competing interests

The authors declare that they have no conflict of interest.

## 8. Acknowledgements

The Levante Canyon Mooring is co-financed by DLTM through funding obtained from Regione Liguria (PAR-FSC 2007-
2013 funds) and by institutional funds from CNR, ENEA, IIM, and INGV. The authors are deeply indebted to the Captain
and the crew of the CNR R/V "Dallaporta" and MM "Leonardo" for continuous support during the whole
measurement/deployment phases.

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
