# Peer review of "Deep water hydrodynamic observations around a Cold-Water Coral habitat in a submarine canyon in the Eastern Ligurian Sea (Mediterranean Sea)"

_Earth System Science Data, 2022_

## Referee Comment (RC1)

The authors present a long-term dataset of hydrological and dynamical observations in the Levante Canyon located in the Eastern Ligurian Sea. These data are particularly precious and relevant as they are collected in a deep canyon, which hosts important and rare marine species. Moreover, these observations can well support sound investigations about deep canyon dynamics. Nevertheless, as the main objective of the paper is the dataset presentation, more information and discussion about meta-data and quality check results must be provided to the reader.

a) Meta information also including sampling interval of each sensor, water column covered by the ADCP, mooring deployment and maintenance … should be resumed in a table;

No information can be found about the sedimentary trap; Time series of ADCP backscatter are also relevant data to be included in the data set;

 b) Fig.3 provide a 2D plot (temporal evolution of profiles) of T and S. The distance between the layers is too high to allow for a reliable interpolation, thus introducing misleading results. Temperature and Salinity are well represented in fig.2 and fig.4. It is a nice representation but I would suggest to eliminate it.

c) Even if data at 88 m depth still present a clear seasonal cycle and inter-annual variability, surface signals are strongly attenuated and filtered out when reaching this depth. Moreover, different dynamic can be found between the surface and this layer, so I would suggest not to assume that they are representative of the upper thermocline, modifying the sentence (paragraph line 203).

d) Salinity data at 350 m depth seem unreasonable for the investigated area and not consistent with the values at the other depths. To this end, I suggest to plot computed density in order to check for the stability of water column or to go back to fig.5 which indicates the LIW denser than waters at higher depths, which is not possible. Even this problem is addressed at the end of ch.4, it still requires a deeper discussion.
As these data were no checked against laboratory salinometer analysis of in situ samples nor compared with CTD profiles and do not seem to satisfy vertical stability, it should be better not to make them available to the scientific community.

e) ADCPs ancillary data provide relevant information about the collected data quality; pressure sensors allow to check for the vertical position of the entire mooring line: add some comments and show something (e.g. show an average profile of the echo intensity to check for the signal attenuation, plot time series of pressure or pitch roll tilt)
Did the other parts of the mooring line affect the ADCP measurements?
How is the correlation between each bin time series?
Are data from the two ADCPs consistent?
Nothing is mentioned about vertical currents, why?

A Northward prevailing current component is evidenced by the reported scatter plots close to the bottom. Add a comment about the role of the topography of the deep canyon in modifying the current direction close to the bottom.

The sampling time resolution of ADCPs (2 h and 1 hours) seems quite low. Two-hours resolution is unable not only to solve the main processes of interest occurring in a canyon (vertical dynamics, internal waves, density currents) but even the analysis of the general circulation features of the region such as tidal and inertial currents cannot be properly addressed. I understand the need to reduce the power consumption to guarantee a longer temporal coverage of the time series, but the objectives of the monitoring and the consequent sampling strategy defined to achieve these objectives must be clear indicated and discussed in the paper.

---

## Author Comment (AC1)

**Author's reply to Comment on essd-2022-466 by Dr. Paola Picco**

The authors present a long-term dataset of hydrological and dynamical observations in the Levante Canyon located in the Eastern Ligurian Sea. These data are particularly precious and relevant as they are collected in a deep canyon, which hosts important and rare marine species. Moreover, these observations can well support sound investigations about deep canyon dynamics. Nevertheless, as the main objective of the paper is the dataset presentation, more information and discussion about meta-data and quality check results must be provided to the reader.

AR: Thank you for taking the time to review our submission and provide us with constructive feedback. Please see below how we have implemented and addressed your suggestions. Specified line numbers refer to the original document.

a) Meta information also including sampling interval of each sensor, water column covered by the ADCP, mooring deployment and maintenance ... should be resumed in a table;

AR: Thank you for this comment. We agree that a resume table is useful to increase the readability of our work. In this regard, a new table (Table 1) and a new paragraph in Section 2.1 has been added to the manuscript, with all the needed meta information.

No information can be found about the sedimentary trap;

AR: Information about the sedimentary trap has been included in Table 1.

Time series of ADCP backscatter are also relevant data to be included in the data set;

AR: Agreed. Time series of ADCP echo intensities are relevant data and have been included in the submitted data set (https://doi.org/10.17882/92236), stored in the NetCDF files. This info has also been added to the manuscript (Lines 130-131).

b) Fig.3 provide a 2D plot (temporal evolution of profiles) of T and S. The distance between the layers is too high to allow for a reliable interpolation, thus introducing misleading results. Temperature and Salinity are well represented in fig.2 and fig.4. It is a nice representation but I would suggest to eliminate it.

AR: Figure 3 was inserted to provide a synthetic view about the dynamics across the water column during the time. The only real interpolation through DIVA was related to temperature measurements recorded at 325 and 350 m by ADCP and CTD respectively. Nevertheless, a revised version of the manuscript without Figure 3 has been provided to avoid confusion.

c) Even if data at 88 m depth still present a clear seasonal cycle and inter-annual variability, surface signals are strongly attenuated and filtered out when reaching this depth. Moreover, different dynamic can be found between the surface and this layer, so I would suggest not to assume that they are representative of the upper thermocline, modifying the sentence (paragraph line 203).

AR: Agreed. The paragraph has been reviewed as suggested.

d) Salinity data at 350 m depth seem unreasonable for the investigated area and not consistent with the values at the other depths. To this end, I suggest to plot computed density in order to check for the stability of water column or to go back to fig.5 which indicates the LIW denser than waters at higher depths, which is not possible. Even this problem is addressed at the end of ch.4, it still requires a deeper discussion. As these data were no checked against laboratory salinometer analysis of in situ samples nor compared with CTD profiles and do not seem to satisfy vertical stability, it should be better not to make them available to the scientific community.

AR: In Figure 5 there was an error related to reported density values for all the three water levels. The following Figure, showing density time-series for the three levels, shows that the vertical stability of the water column is indeed satisfied.

[Figure]

Nevertheless, as density and salinity values recorded by the CTD at 350 m depth are high for the investigated area we decide to flag them as potentially correctable bad data in the re-submitted data set (https://doi.org/10.17882/92236). In order to fix this, future measurements were already planned to be checked against a laboratory salinometer and compared with concomitant CTD profiles.

e) ADCPs ancillary data provide relevant information about the collected data quality; pressure sensors allow to check for the vertical position of the entire mooring line: add some comments and show something (e.g. show an average profile of the echo intensity to check for the signal attenuation, plot time series of pressure or pitch roll tilt)

AR: Agreed. As a fact, the QC procedure was done taking this into account. It's not only the pressure sensor, it's a series of controls and cross-validations that are presented within the manuscript. As regards the RDI, the manuscript has been revised providing a pitch/roll/tilt figure. For the Nortek the QC procedure related to data with tilt effect, has been performed throughout the Nortek Surge post-processing software.

Did the other parts of the mooring line affect the ADCP measurements?

AR: The mooring layout has been designed to avoid interference between sensors and to minimize disturbance caused by the mooring itself. Furthermore, in the case of ADCP measurements affected by the mooring inclination, the measurements have been flagged by the appropriate flags as results of QC procedure.

How is the correlation between each bin time series? Are data from the two ADCPs consistent?

AR: The RDI NetCDF files contain the pulse-to-pulse correlation in a ping for each depth cell. As a decrease in the correlation is a decrease in data accuracy, measurements with low correlation cannot pass the quality control and are flagged with a number different than 1. The threshold of the minimum accepted correlation value is depending on the ADCP model and the transmitted frequency. RDI, for example, recommends a threshold value of 64 for the Workhorse Long Ranger 75 kHz model and 120 for the Ocean Observer 38 kHz. In our case, a threshold of 110 was used, as suggested by the manufacturer. The NORTEK NetCDF does not contain the correlation values as the quality control was done by the use of the NORTEK-provided software SURGE. A complete correlation analysis is out of the scope of the present preliminary publication. As soon as additional long term time-series will be available for the site, we will certainly add more results for this topic in a future publication intended for a dedicated oceanographic Journal.

Nothing is mentioned about vertical currents, why?

AR: Vertical currents have been disregarded because their magnitudes are typically of the same order of accuracy of both the ADCPs. Time series of RDI ADCP vertical velocities have been included in the submitted data set (https://doi.org/10.17882/92236), stored in the NetCDF files.

A Northward prevailing current component is evidenced by the reported scatter plots close to the bottom. Add a comment about the role of the topography of the deep canyon in modifying the current direction close to the bottom.

AR: Agreed. A comment in this sense was already reported at L265-266, and a new paragraph has been added thereafter as suggested.

The sampling time resolution of ADCPs (2 h and 1 hours) seems quite low. Two-hours resolution is unable not only to solve the main processes of interest occurring in a canyon (vertical dynamics, internal waves, density currents) but even the analysis of the general circulation features of the region such as tidal and inertial currents cannot be properly addressed. I understand the need to reduce the power consumption to guarantee a longer temporal coverage of the time series, but the objectives of the monitoring and the consequent sampling strategy defined to achieve these objectives must be clear indicated and discussed in the paper.

AR: Unfortunately, there was a typing error: the sampling time resolution is 1h for both the RDI and the Nortek ADCPs. The paper has been revised accordingly (Line 121). This information has been also added in Table 1. This resolution is considered acceptable to solve the inertial signal at our latitudes where with measurements every 1h, we would have about 20 points for each cycle. During this first period the priority was to understand the long-term hydrological dynamics along the water column in the area and hence to reduce the power consumption to guarantee a longer temporal coverage of the time series. Here the main goal of the paper is the presentation of the long-term dataset in such a peculiar environment, which hosts vulnerable marine species such as the CWC that we identified in previous investigations.

---

## Author Comment (AC2)

**Author's reply to Comment on essd-2022-466 by Anonymous Referee #2**

**General Comments**

This study presents some high-quality high-resolution physical oceanographic data in an interesting area in the Mediterranean Sea that is not highly accessible and hosts a cold water coral community. Given that the data collection will continue and future data be accessible to the scientific community, it certainly meets the ESSD criteria for data availability. The presentation of the instruments, data collection, processing as well as quality control is very detailed and I believe in the quality of the data (although I may not have the most appropriated expertise to judge that).

AR: We really thank the reviewer for taking the time to review this manuscript, and for these positive comments and statements about the fulfilment of the ESSD criteria. Please see below how we have implemented and addressed your suggestions. Specified line numbers refer to the original document.

However, I end up reading through the manuscript feeling unsatisfactory in certain aspects. I agreed to review this manuscript mainly because my personal research interest in cold water corals. This manuscript is instead mainly a technical report on the physical oceanographic data only, without presentation of the biogeochemical or sediment trap data. While I understand the additional data collection and processing may take more time, I do not see a logical plan from the authors as to how these additional data can be integrated into the current physical oceanographic data. Even if the biogeochemical and sediment trap data are not ready, I would at least like to see in what aspects the current physical oceanographic data can help up better monitor the growth environment of the coral community. I think the dataset will have higher impacts in the scientific community and the general public if these points can be addressed.

AR: Dear referee, thanks for your comments, we understand and see your point. In this paper we present a long-term dataset of hydrological and dynamical observations in the Levante Canyon, in the Eastern Ligurian Sea. We believe this is the first important step to understand the hydrological dynamics along the water column in this area. Certainly those data are particularly relevant as they are collected in a deep canyon, which hosts vulnerable marine species such as the CWC that we identified in previous investigations. Here the main goal of the paper is the presentation of the long-term dataset in such a peculiar environment. We believe these observations can well support sound investigations about the deep canyon dynamics and consequently the CWC habitat conditions (for example the current velocity in the proximity of the sea bottom). Shortly we'll proceed with a second publication dedicated to the sediment trap data collected on this mooring station, with the focus on the sediment input to the sea bottom morphology and the corals growing conditions related to the sediment input.

**Specific Scientific Comments**

Line 114 & 119: Just out of curiosity, why is the precision for the two temperature sensors on the ADCPs so different? Also 0.4°C uncertainty seems pretty large for temperature measurements.

AR: The reported precision value for the RDI QuarterMaster ADCP is correct. ADCPs use the temperature at its transducer to calculate sound speed and its temperature values should be only taken into account as environmental monitoring data if quality controlled against reference data. In our manuscript temperatures recorded by the RDI have been considered reasonable as comparable to the

closest CTD measurements. Despite its smaller uncertainty, temperature data recorded by the NORTEK ADCP were instead considered not reliable after the analogous quality control check.

Line 180: For this paragraph, I am not sure about how the authors are integrating sediment trap and geochemical data with the physical oceanography data given the current description. I think a more logical description of how all these different datasets are related and will be used together is needed.

AR: We have modified the manuscript there, at the beginning of the Results paragraph, according to your comments, thanks.

So we clarify that the hydrodynamic data presented here constitute the fundamental basis for a deep understanding of the ecological conditions that allow the hotspot of deep water corals along the Levante Canyon. For the first time we present here a 2 years data of temperature, salinity and water currents along the water column. So we understand the seasonal variability of hydrological conditions, the temperature range and so on. With a future work, we will focus on the sediment input that arrives down at the sea bottom of the Canyon (thanks to the sediment trap at 582 m depth) where CWC live and we'll understand the interactions among oceanography, sediment input, biogeochemistry and spatial distribution of CWC biological communities, for a deep understanding of the Levante Canyon system functioning.

Figure 5: I am not sure what all the different symbols mean for this plot.

AR: Noted. Please consider that the Figure has been revised as it was not correct as regards sigma values (i.e. water density anomalies).

Line 262: I am not sure what "interest" means here

AR: The word "interest" has been replaced by "affect" in the new version of the manuscript.

Line 315: This is one of the more interesting observations that the effect of the heat wave was not significant at the station. I feel like it warrants a slightly more detailed analysis as to why that is, given the current knowledge about the circulation patterns in the area, although I understand the ESSD journal is more focused on the presentation of the data.

AR: Agreed, this result is really interesting but out of the scope of the ESSD journal. As soon as additional long term time-series will be available for the site, we will certainly add more results for this topic in a future publication intended for a dedicated oceanographic Journal.

**Technical Comments**

AR: All text corrections about Technical Comments have been edited as detailed below.

Line 14 and 18: "2-year" instead of "2 years"

AR: Corrected.

Line 63: "Panel (c)" instead of "the panel (c)"

AR: Corrected.

Line 66: "generate" instead of "originate"

AR: Corrected.

Line 69–72: This is an incomplete sentence without a verb. Either rewrite it, or connect it with the previous sentence with a colon.

AR: Corrected adding a colon.

Line 83: "temporal-spatial" instead of "time-spatial"

AR: Done.

Line 91 & 142: "first" instead of "firstly"

AR: Done.

Line 128: remove "respectively"

AR: Done.

Table 1: In the rows corresponding to Flags 2 and 4, the description sentences are broken in two lines awkwardly

AR: Corrected. Please note that we have also removed coloured cells from the Table as they were not suitable for HTML conversion of the paper, as raised by the editorial support team.

Line 179: "presented" instead of "proposed", "shows" instead of "is well shown as well"

AR: Done.

Line 181: "trap" instead of "traps"

AR: Done.

Line 223: "With regards to" instead of "As regards"

AR: Done.

Line 278: I don't know what "NC" means here and cannot find previous mentions of the abbreviation.

AR: NC stands for Northern Current as mentioned at Line 70. The mention has been repeated in the revision to enhance the readiness.

Line 318: "reached" instead of "was reached"

AR: Done.

---

## Author Response (AR2)

**Author's reply (AR) to Comment on essd-2022-466 by the Topical Editor (TE)**

TE: The reported 2-year hydrodynamic dataset is from a unique deep-sea coral habitat and might help protect this vulnerable ecosystem under climate change. In addition, the data might be useful to the paleoceanography community. It is important that the authors continue to update this dataset when more data becomes available.

AR: Thank you for taking the time to review our submission and provide us with such constructive feedback. We certainly confirm our interest and willingness in maintaining the Levante Canyon Mooring in the long term and in updating the scientific community, sharing future data and results.

Please consider that few typos and some minor inaccuracies have been corrected in the manuscript.

Figures have been revised to improve their quality, in particular as regards font dimension and marker size (i.e., Figure 2, 6, 7, 8, 9 and 10). Figure 3, Figure 4 and Figure 5 have been corrected as regards the labeling of X- and Y-axis. In Figure 7, the colors have been changed in accordance with Figure 8, 9 and 10 and the velocity range has been reduced from 0 to 0.5 m/s. Higher-resolution figures have been uploaded.

In Table 3 and 4 some repetitions in the unit of measurement have been deleted.